# The Contribution of the Pore Size of Titanium DC (Direct Current) Sputtered Condensation Polymer Materials to Electromagnetic Interruption and Thermal Properties

Hye-Ree Han

Department of Beauty Art Care, Graduate School of Dongguk University, Seoul 04620, Republic of Korea; luckyherry@daum.net

**Abstract:** Using special materials has been in the spotlight, along with their multifunctional demands, research on electromagnetic interruption, thermal characteristics, biosignal sensors, secondary batteries, etc. In this study, titanium was sputtered into a condensation polymer material and considered in depth in terms of electromagnetic interruption, thermal properties, infrared blocking, etc. As a result of observing the electromagnetic wave shielding effect, the electromagnetic wavelength value decreased from 168.0 to 42.7 to 64.0 when titanium DC sputtered film samples were placed in front of the electromagnetic wave source. The titanium DC sputtered samples significantly reduced electrical resistance compared to the untreated samples. In addition, the IR transmittances of the titanium sputtered specimens were decreased compared to the untreated specimens. When only the cross-section was treated with titanium sputtering and the titanium surface was directed toward the infrared irradiator, the infrared permeability was 64.3 to 0.0%. After taking an infrared thermal image, $\Delta H$, $\Delta V$, $\Delta S$, $\Delta Y$, $\Delta Cr$, and $\Delta Cb$ values were calculated. It is believed that the titanium DC sputtered polyamide materials produced in this study can be used for high-functional protective clothing, sensors by applying electromagnetic interruption, IR blocking, and stealth functions.

**Keywords:** electromagnetic interruption; DC sputtering; titanium





## 1. Introduction

Titanium (Ti) has a melting point of 1668 °C, higher than that of iron, and has superior corrosion resistance, low thermal expansion heat conductivity, so it has less deformation, and abundant reserves. In addition, titanium is a highly reactive metal, so it reacts with oxygen in the air and forms a titanium oxide film. Titanium has strength and ductility, and is used in various fields, such as electrolytic electrodes, nuclear power generation service devices, corn rods, artificial bones, surgical tools, seawater desalination devices, swivel parts, shafts, LNG seawater cooler tubes, and various mechanical parts.

There are various studies on laser treatment for mechanical stress, the atomic layer deposition of chlorine-containing titanium-zinc oxide nanofilms, 3D-printed titanium lattice structure, titanium alloys, titanium surface modification, porous titanium, and antibacterial applications [1–12].

In addition, sputtering technologies are an environmentally friendly manufacturing technique that does not produce wastewater. This technology also has the advantage of conveniently coating various metals on samples, so it is applied in various ways to high-tech mechanical parts and semiconductors. There are studies on electrical properties, physical properties, thermal stability, biosignal sensors, and hydrophobic thin films using sputtering technology [13–21].

In addition, various studies on the electrical, magnetic, chemical, physical, and sensing characteristics of composite materials related to electrical conductivity are being conducted [22–28].

Regarding infrared rays, research on infrared image fusion, object-oriented attention, infrared and visible light image fusion, and remote sensing images is also being actively conducted [29–32]. The aim of infrared and visible image fusion techniques is to extract and integrate features from images captured using various sensors using specific algorithms to create complementary images that contain both rich detailed features of visible images and target information of infrared images. The technology has far-reaching implications for economic development, including applications in areas such as defense, military, intelligent transportation, and power grid operations [29]. Lv et al. conducted a comprehensive experiment on two infrared datasets, and the results show the efficiency of our approach. We propose a method of generating captions for infrared images based on object-oriented attention. Our approach includes two models: detector and LSTM. The approach of this study is suitable for deployment on embedded devices because it requires fewer resources and is easy to deploy [30]. Xiaodi Xu and others have important applications for the fusion of infrared and visible images in various engineering fields. However, in the current fusion of infrared and visible images, the texture details of the fused images are unclear, and the infrared targets and texture details are disproportionately displayed, resulting in information loss. In this paper, we propose an improved Generative Adversarial Network (GAN) fusion model for the fusion of infrared and visible images [31]. Zhao et al. propose a CNN-based layer adaptive GCP extraction method for TIR RSI. Specifically, the built feature extraction network consists of a primary module and a layer-adaptive module [32].

In addition, there are many studies that have introduced artificial intelligence, electric conductivity, composite multi-function materials, and composite porous materials. This research aims to study, in depth, infrared function, electrical properties, etc., of the titanium sputter technique. In addition, the titanium sputter technique was used to coat the surface for longer than in previous studies to cover the metal grain layer thickly. Moreover, this study aims to examine the correlation between IR images and infrared transmittance. Based on the results of the research data, the applicability to industrial fields is considered.

In this paper, the electromagnetic wave blocking performance of titanium sputtering-treated polyamide materials, stealth effects on infrared cameras, electrical conductivity, transmittance of IR, electromagnetic wave blocking properties, and thermal properties were examined using various methods. In addition, a titanium sputtered samples were prepared by changing the pore size of the polyamide sample.

Therefore, titanium sputtering was performed by dividing polyamide into films, plain fabrics, and mesh (steps 1 to 5) to observe various characteristics according to sample pore size. Furthermore, based on the study results, electromagnetic blocking performance, heat transfer characteristics, stealth effects on IR cameras, and the applicability of high-functional smart materials were considered.

## 2. Materials and Methods

The materials used for titanium DC sputtering treatment are polyamide materials (film, fabric, and mesh, steps 1–5). Titanium sputter process was prepared with various pore sizes. In addition, the sample characteristics are as shown in Table 1.

**Table 1.** Information of base polyamide materials.

|  | Poly Amide Film | Poly Amide Fabric | Poly Amide Net 1 | Poly Amide Net 2 | Poly Amide Net 3 | Poly Amide Net 4 | Poly Amide Net 5 |
|---|---|---|---|---|---|---|---|
| Sample code | TF1 | TA1 | TN1 | TN2 | TN3 | TN4 | TN5 |
| Sample thickness (mm) | 0.09 | 0.15 | 0.08 | 0.10 | 0.15 | 0.11 | 0.19 |
| Weave type | Film | Plain weave | Plain mesh | Plain mesh | Plain mesh | Plain mesh | Plain mesh |
| Pore size ($\mu m^2$) | 0 | 1200 | 3564 | 35,696.3 | 94,440.4 | 136,476.2 | 294,825.5 |

The titanium sputtering process applied to the polyamide materials were as represented in Table 2. Sputter coater (SORONA, SRN-120, Pyeongtaek, Korea) is the device used

for titanium sputtering treatment. When the titanium sputter technique was performed, the base polyamide material diameter of the circular shape was 19.5 cm. In this study, the pore size was adjusted by changing the density of warp and weft yarns during weaving. Sputtering is a type of vacuum deposition method commonly used in integrated circuit production line processes, which accelerates gas, such as ionized argon, at a relatively low vacuum degree, collides with a target, and ejects atoms to create a film on a substrate such as a wafer or glass.

**Table 2.** Titanium sputter conditions.

| | |
|---|---|
| Time (s) | 2400 |
| Process pressure (Torr) | 6 m |
| Gas (sccm) | Ar 40 |
| Power (W) | DC 600 |
| Machine | SRN-120 |

As a result of FE-SEM with a digital microscope after titanium sputter process, it was revealed that titanium was well formed on the polyamide surface in all sputtered specimens.

In the case of electromagnetic wave blocking performance evaluation, an electromagnetic-wave-measuring device (9024-EN-00, Electromagnetic radiation tester, China) was used to place a sample between the electromagnetic wave generator and the electromagnetic-wave-measuring device to measure the degree of electromagnetic wave blocking.

Infrared intensity tester (5 mm Infrared LED, Infrared Emitting Diodes, EVERLIGHT) was utilized for IR transmittance. The IR strength irradiated to the specimens was $200 \text{ W/m}^2$. In addition, for major infrared wavelength, it was 940 nm.

For stealth characteristics, IR Thermographic camera (Fliri7) was used. The specimen was placed close to the heat source and a thermal images were obtained using an IR thermal camera. The hidden effect and values of H, V, S, Y, Cr, and Cb were recorded for infrared thermal imaging cameras utilizing the Color Inspector 3D program. In addition, more quantitative color differences were studied for H, S, V, Y, Cb, and Cr data.

In addition, ΔH, ΔV, ΔS, ΔY, ΔCr, and ΔCb data were calculated. The data of ΔH, ΔS, and ΔV values are expressed in the following Equations (1)–(3).

$$\Delta H = H\_treated - H\_untreated \tag{1}$$

$$\Delta S = S\_treated - S\_untreated \tag{2}$$

$$\Delta V = V\_treated - V\_untreated \tag{3}$$

in the definitions of Equations (1)–(3) are as follows.

H_untreated: data H of untreated samples;
H_treated: data H of sputtered samples;
S_untreated: data S of the untreated samples;
S_treated: data S of the sputtered samples;
V_untreated: data V of the untreated samples;
V_treated: data V of the sputtered samples.

Equation (1) shows the degree of change in the "H" data due to titanium sputtering treatment by subtracting the "H" data of the untreated specimen from the "H" data of the titanium sputtered sample. The value "H" in the concept of color space means a color angle (hue). In the Equation (2), the change in the "S" data due to titanium sputter technique is the data obtained by subtracting the "S" data of the "untreated fabric" from the "S" data of the titanium sputtered sample. The "S" value in the color space represents the concentration and saturation of the color. Equation (3) demonstrates that in this case, the "V" data of the untreated specimen are subtracted from the "V" data of the titanium sputtered sample, indicating the color difference caused by titanium sputtering technique. The "V" value in the color space represents the degree of brightness and value. Using the

"T" data of Equation (7), the color difference between the titanium sputtered specimen and the untreated specimen in color space was calculated.

$\Delta Y$, $\Delta Cb$, and $\Delta Cr$ values were calculated as follows.

$$\Delta Y = Y\_treated - Y\_untreated \tag{4}$$

$$\Delta Cb = Cb\_treated - Cb\_untreated \tag{5}$$

$$\Delta Cr = Cr\_treated - Cr\_untreated \tag{6}$$

$$\Delta T = \sqrt{(\Delta Y)^2 + (\Delta Cb)^2 + (\Delta Cr)^2} \tag{7}$$

Definitions of Equations (4)–(7) are provided below.
Y_untreated: data Y of untreated specimen;
Y_treated: data Y of sputtered specimen;
Cb_untreated: data Cb of the untreated specimen;
Cb_treated: data Cb of the sputtered specimen;
Cr_untreated: data Cr of the untreated specimen;
Cr_treated: data Cr of the sputtered specimen.

Equation (4) shows a change in the "Y" data due to titanium sputtering technique by subtracting the "Y" data of the untreated fabric from the "Y" data of the titanium sputtering treatment sample. Equation (5) shows the change in the data of "Cb" due to titanium sputtering technique by subtracting the data of "Cb" of "untreated sample" from the data of "Cb" of the titanium sputtering specimens. Equation (6) shows the change in Cr data by titanium sputter process by subtracting the data "Cr" of the untreated specimens from the data "Cr" of the titanium sputtered specimens.

## 3. Results and Discussions

### 3.1. Surface Geometry

FE-SEM photography images were captured to observe the nano-shape formed on the titanium sputtering-treated sample (Figure 1). As a result of observing the micrograph, it was possible to capture that a titanium layer was formed on the surface of all titanium sputtering-treated samples. In addition, that a titanium surface in the form of a meteorite was formed in all samples could be captured. Meteorite-shaped titanium sizes were captured at 45.58–80.98 and 53.63–67.24 nm for TF1 and TA1, respectively. In the case of TN1, TN2, TN3, and TN4, titanium sizes were 83.37 to 117.2, 95.24 to 100.1, 62.19 to 118.0, and 55.15 to 79.60 nm, respectively. In previous studies, after sputtering treatment, FE-SEM image observations showed that nanograins were formed on the surface [33], and in this research, they showed a meteorite shape rather than a grain shape on the surface. Since the sputtering process time was longer than that of previous studies, it was judged that the titanium layer was more coated, resulting in an irregular meteorite shape. Moreover, the result of FE-SEM prompts research in more detail on what the thickness of the titanium layer is on the specimens in Figure 2 is equal to. In the case of TF1 and TA1, the thickness of the titanium layer was captured as 718.5 to 751.4 and 470.8 to 521.7 nm, respectively. In the case of TN1, TN2, TN3, and TN4, the thickness of the titanium layer was 606.0 to 619.7, 553.9 to 578.6, 693 to 803.5, and 551.2 to 567.6 nm, respectively. The EDX data are as shown in Figure 3, which proved that titanium materials were well coated on the sample surface.

### 3.2. Characteristics of Electromagnetic Interference

The electronic wave blocking performance of the titanium sputtered specimens is considered to be related to the pore size. As a result of observing the electromagnetic wave shielding effect, the electromagnetic wavelength (V/m) value decreased from 168.0 to 42.7 to 64.0 when titanium sputtering-treated film samples were placed in front of the electromagnetic wave source (Figure 4). In addition, the titanium coating layer was

measured to block electromagnetic waves better when it was facing away from electromagnetic irradiator than when facing the electromagnetic irradiator (titanium phase down, 42.7 V/m). When titanium sputtering was performed on other polyamide mesh, it was confirmed that there was little electromagnetic wave blocking effect. The smaller the pore size (294,825.5 → 0 μm²), the better the electromagnetic wave blocking performance. In previous studies with aluminum sputtered specimens, as the pore size decreased, the electromagnetic wave transmission values were from 25.7 mG to 49.0 mG, and the electromagnetic wave blocking performance decreased [33]. It is believed that electromagnetic waves pass between the pores using the pores between the samples. However, in the case of films, it is thought that the titanium layer blocks electromagnetic waves because there are no pores. However, in order to more completely block electromagnetic waves so that the V/m value becomes 0, it is believed that the titanium layer must be sputtered thicker on the film.

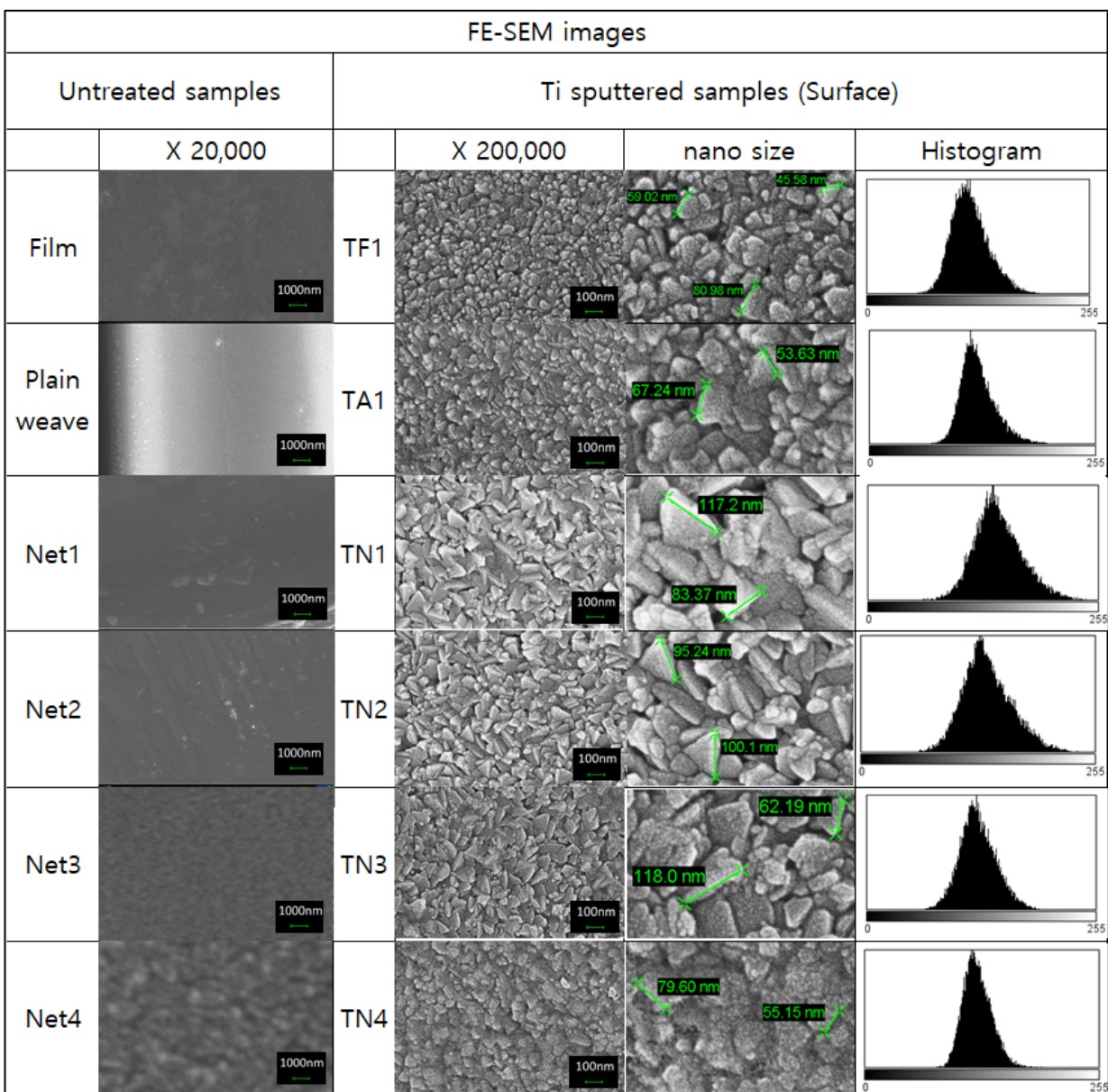

**Figure 1.** Surface characteristics of titanium DC sputtered polyamide materials.

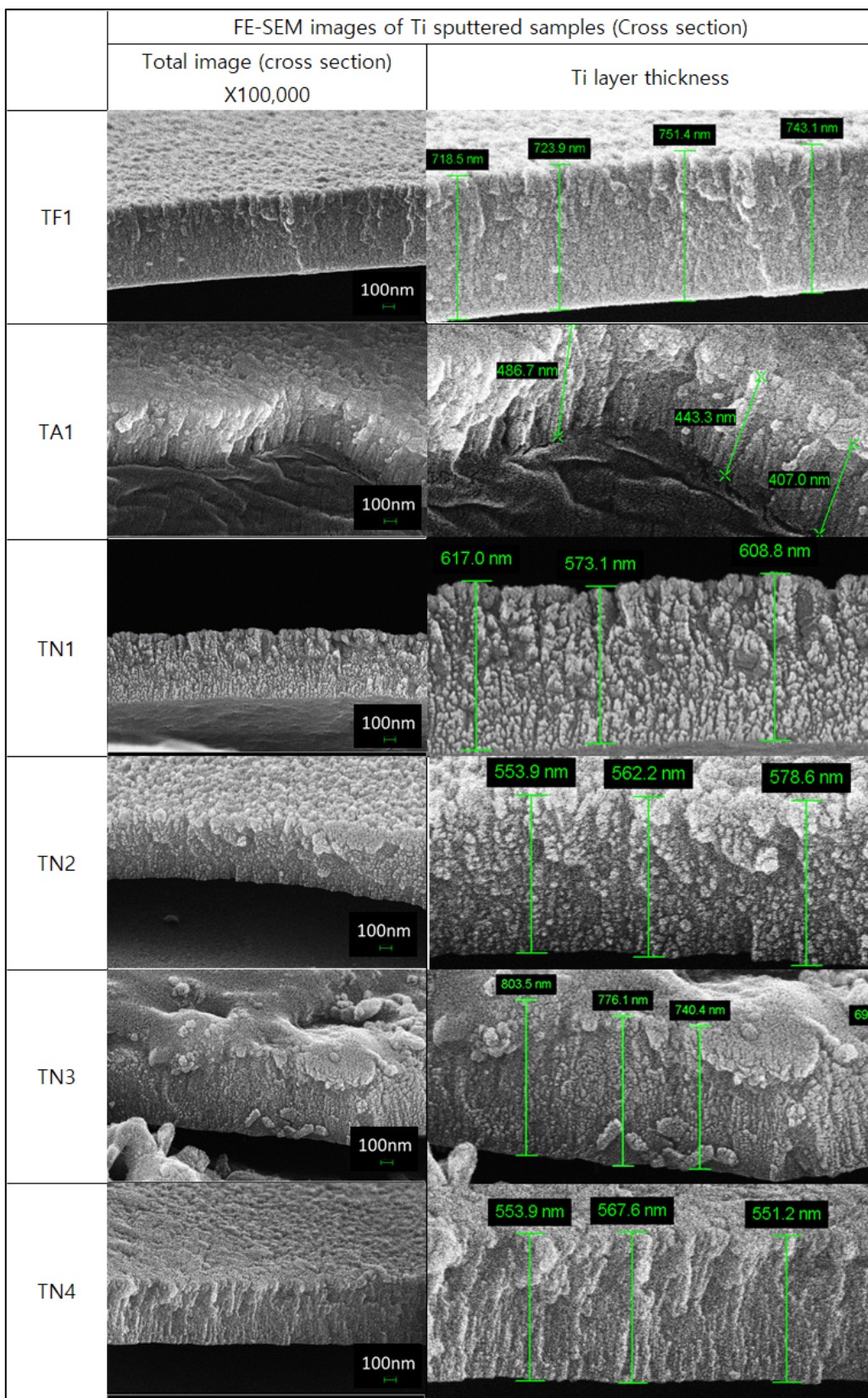

**Figure 2.** Cross-section characteristics of titanium DC sputtered polyamide materials.

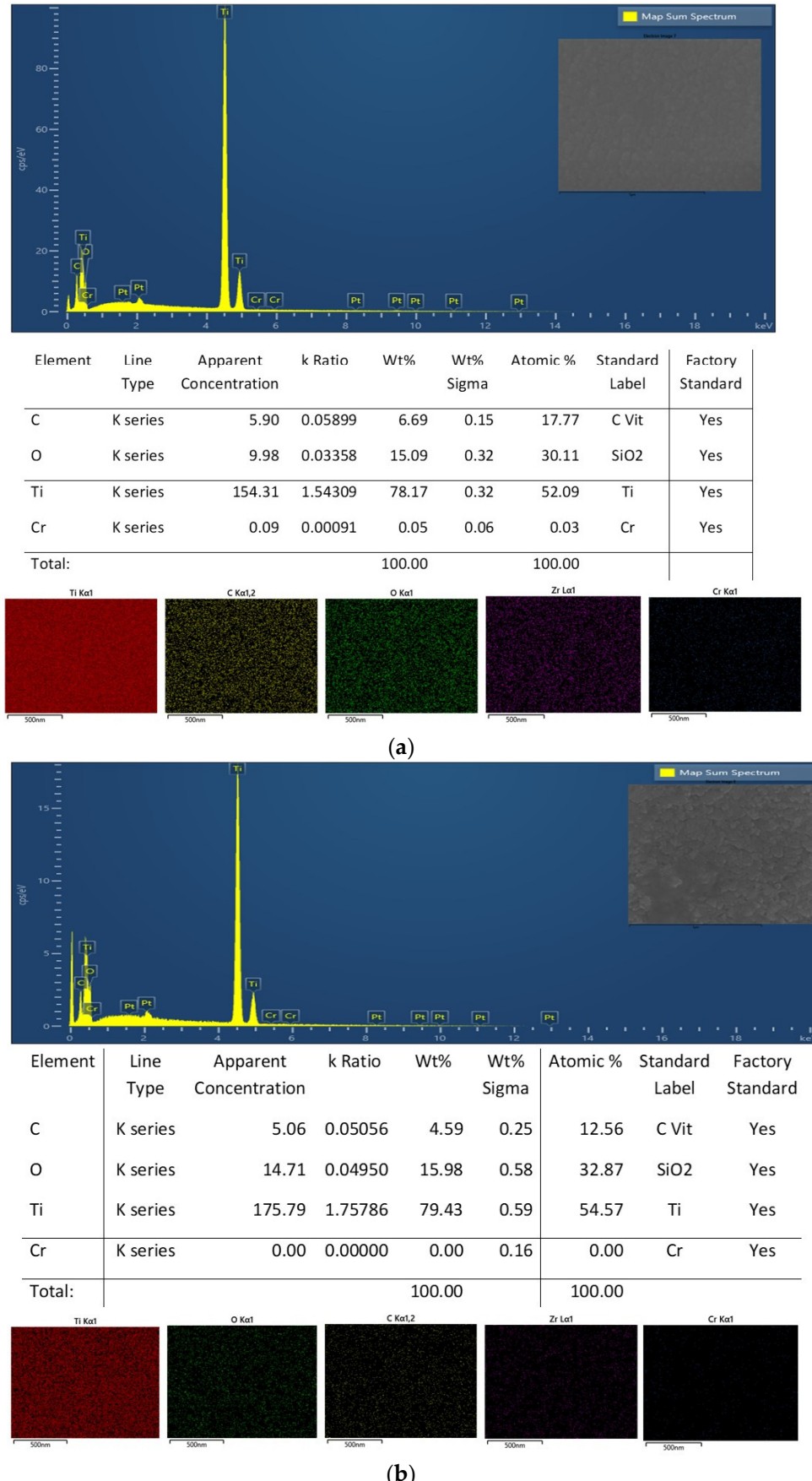

| Element | Line Type | Apparent Concentration | k Ratio | Wt% | Wt% Sigma | Atomic % | Standard Label | Factory Standard |
|---|---|---|---|---|---|---|---|---|
| C | K series | 5.90 | 0.05899 | 6.69 | 0.15 | 17.77 | C Vit | Yes |
| O | K series | 9.98 | 0.03358 | 15.09 | 0.32 | 30.11 | SiO2 | Yes |
| Ti | K series | 154.31 | 1.54309 | 78.17 | 0.32 | 52.09 | Ti | Yes |
| Cr | K series | 0.09 | 0.00091 | 0.05 | 0.06 | 0.03 | Cr | Yes |
| Total: | | | | 100.00 | | 100.00 | | |

(a)

| Element | Line Type | Apparent Concentration | k Ratio | Wt% | Wt% Sigma | Atomic % | Standard Label | Factory Standard |
|---|---|---|---|---|---|---|---|---|
| C | K series | 5.06 | 0.05056 | 4.59 | 0.25 | 12.56 | C Vit | Yes |
| O | K series | 14.71 | 0.04950 | 15.98 | 0.58 | 32.87 | SiO2 | Yes |
| Ti | K series | 175.79 | 1.75786 | 79.43 | 0.59 | 54.57 | Ti | Yes |
| Cr | K series | 0.00 | 0.00000 | 0.00 | 0.16 | 0.00 | Cr | Yes |
| Total: | | | | 100.00 | | 100.00 | | |

(b)

**Figure 3.** *Cont.*

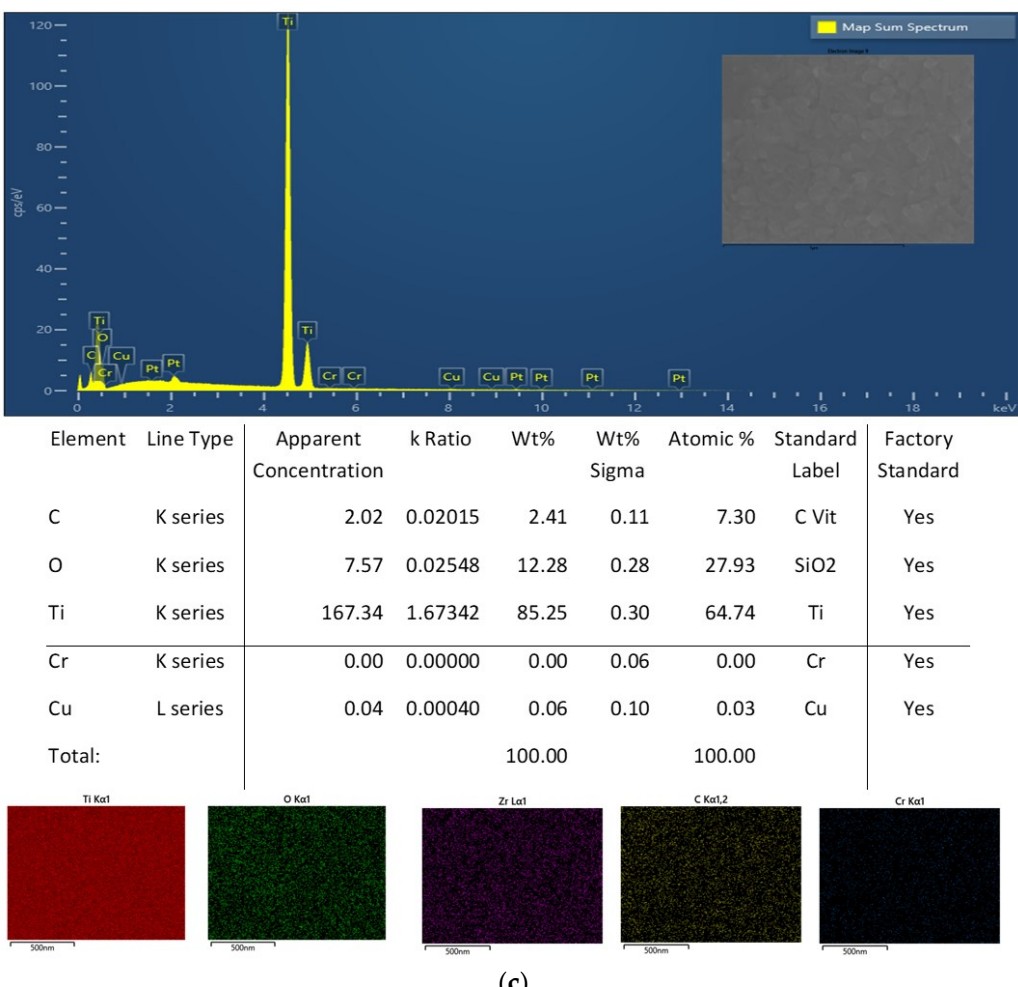

| Element | Line Type | Apparent Concentration | k Ratio | Wt% | Wt% Sigma | Atomic % | Standard Label | Factory Standard |
|---------|-----------|------------------------|---------|------|-----------|----------|----------------|------------------|
| C | K series | 2.02 | 0.02015 | 2.41 | 0.11 | 7.30 | C Vit | Yes |
| O | K series | 7.57 | 0.02548 | 12.28 | 0.28 | 27.93 | SiO2 | Yes |
| Ti | K series | 167.34 | 1.67342 | 85.25 | 0.30 | 64.74 | Ti | Yes |
| Cr | K series | 0.00 | 0.00000 | 0.00 | 0.06 | 0.00 | Cr | Yes |
| Cu | L series | 0.04 | 0.00040 | 0.06 | 0.10 | 0.03 | Cu | Yes |
| Total: | | | | 100.00 | | 100.00 | | |

(**c**)

**Figure 3.** EDX results of molybdenum sputtered polyamide specimens: (**a**) TF1, (**b**) TA1, and (**c**) TN1.

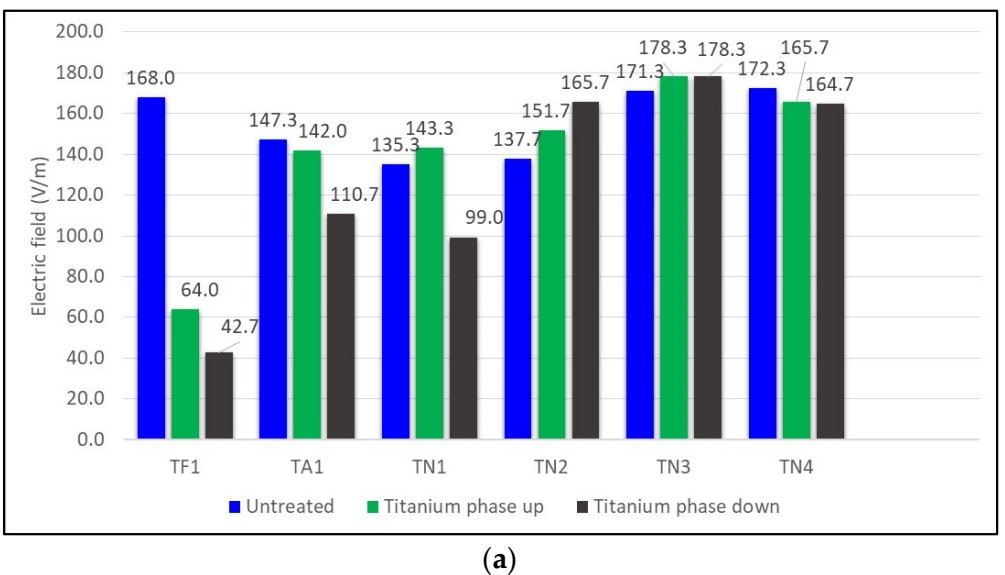

(**a**)

**Figure 4.** *Cont.*

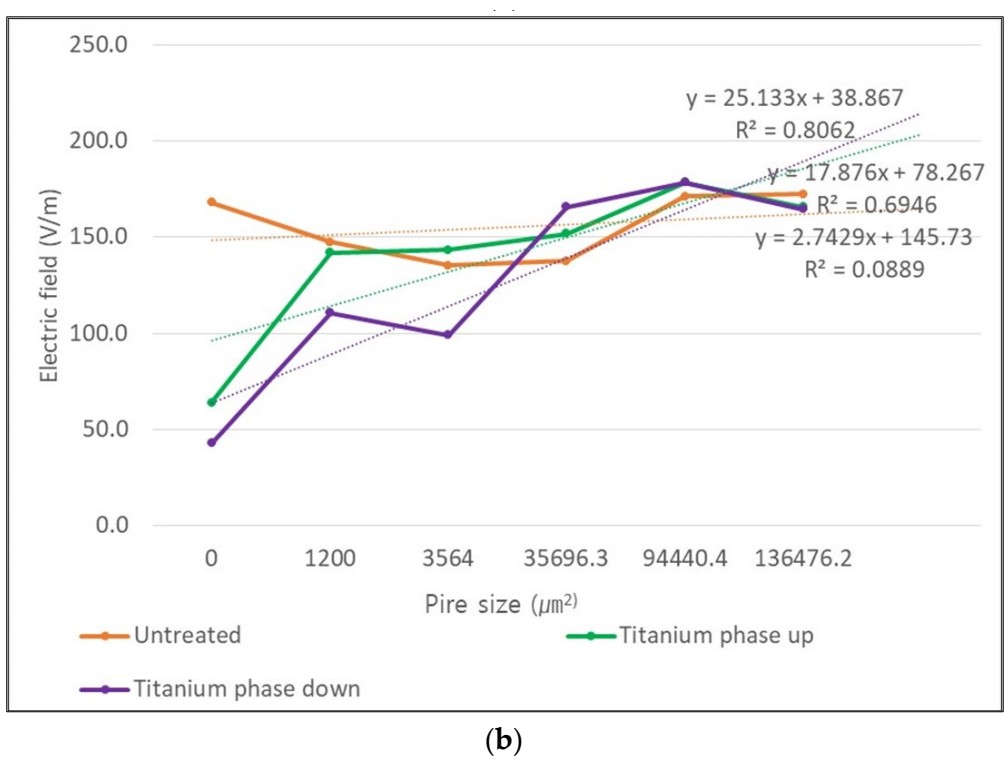

**(b)**

**Figure 4.** Characteristics of electric field of specimens. (**a**) Electromagnetic interruption of titanium DC sputtered polyamide specimens. (**b**) Regression line of electric field.

### 3.3. Characteristics of Electrical Conductivity

The results of the electrical conductivity of titanium sputtered specimens are considered to be related to structure and pore size. Compared to the untreated specimens, the electrical resistance data of titanium sputtered film and mesh specimen were decreased (Figure 5). In the results of the untreated specimens, the electrical resistance of TF1, TA1, and TN1–5 all showed "over load". However, after the titanium sputtering process, the electrical resistance values of all specimens other than NFa were significantly reduced. In this research, the sputtering time of the titanium sputtering-treated TN1 to TN5 samples is long, so the thickness of the titanium coating layer is thick, and the current is not cut off and electrical conductivity is reduced. However, in the case of the TA1 sample, the electrical conductivity was relatively high, even when the metal sputtering process was performed under the same conditions. This is believed to be due to the dense intersection of weave yarn, which has many deadlock points, and the height of the weave yarn furrow is higher than that of the mesh. Previous studies using aluminum sputtering samples showed a similar trend: films showed good electrical conductivity, while fabrics did not. It was explained that this is judged to be due to the fact that the metal layer is cut off due to the fraying of the warp and weft yarns, and the lack of current flow [33].

### 3.4. Infrared Transmittance Behavior

In this paper, infrared transmittance experiments of untreated samples and cross-section titanium sputtered samples were conducted. The experimental data are represented in Figure 6. An infrared irradiator was placed on the left, and infrared measuring apparatus was placed on the right. In addition, the titanium sputtered specimens were placed between the irradiator equipment and the measuring apparatus. As a result of the experiment, the sample treated with titanium sputter treatment showed a significant reduction in IR transmission data.

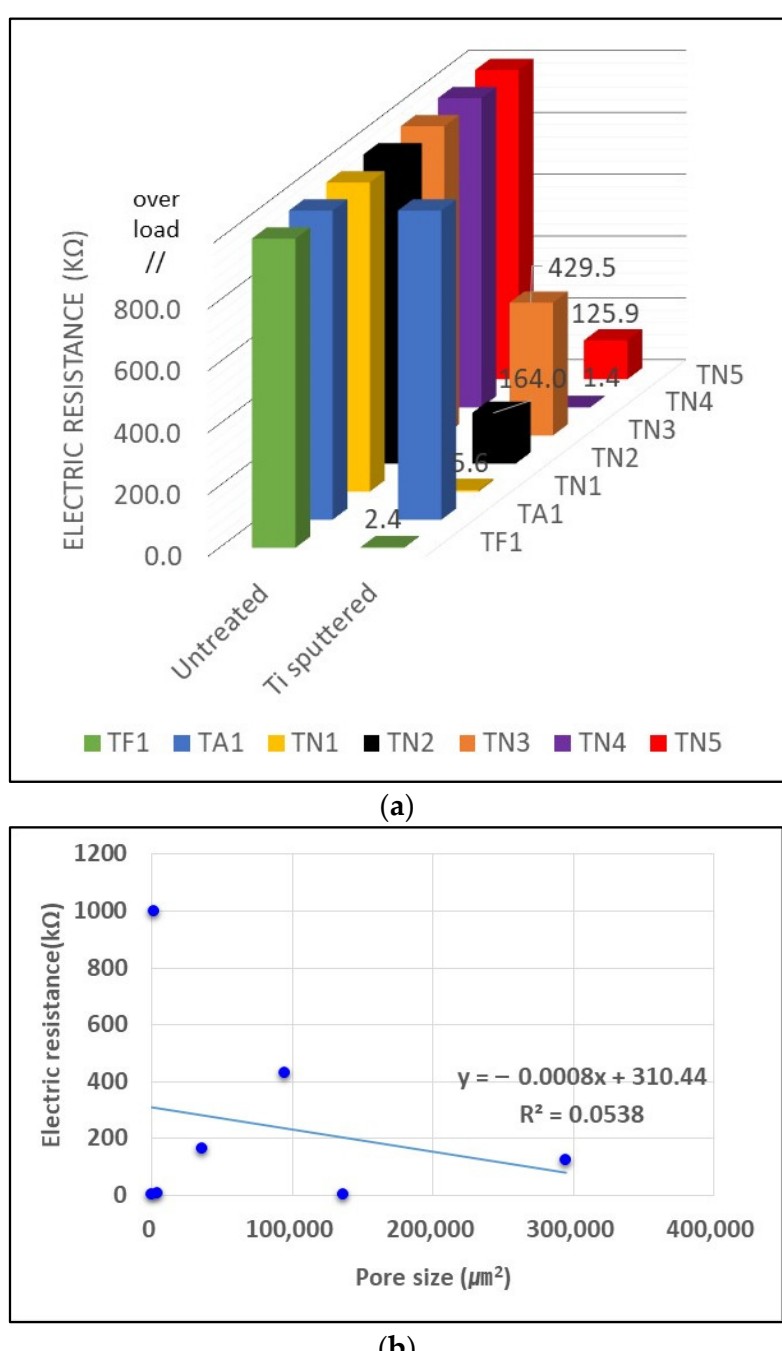

**Figure 5.** Characteristics of electrical conductivity of specimens. (**a**) Electrical resistance of specimens. (**b**) Correlation between electrical resistance and pore size.

In addition, when titanium sputtering was performed on the cross-section and the titanium surface faced the infrared measuring apparatus (titanium phase: back), the infrared transmittance was 67.2%–0.0%. Namely, the changes in the infrared transmission data according to the direction of the titanium sputtered layer were not large. However, However, when the titanium surface was directed to the infrared irradiator, the IR transmittance data was slightly reduced compared to when the titanium surface was directed to the infrared measuring apparatus. In this research, since the thickness of the titanium layer was made thicker than in past studies, it is judged that the difference in infrared transmittance (%) according to the direction of the titanium surface was very small.

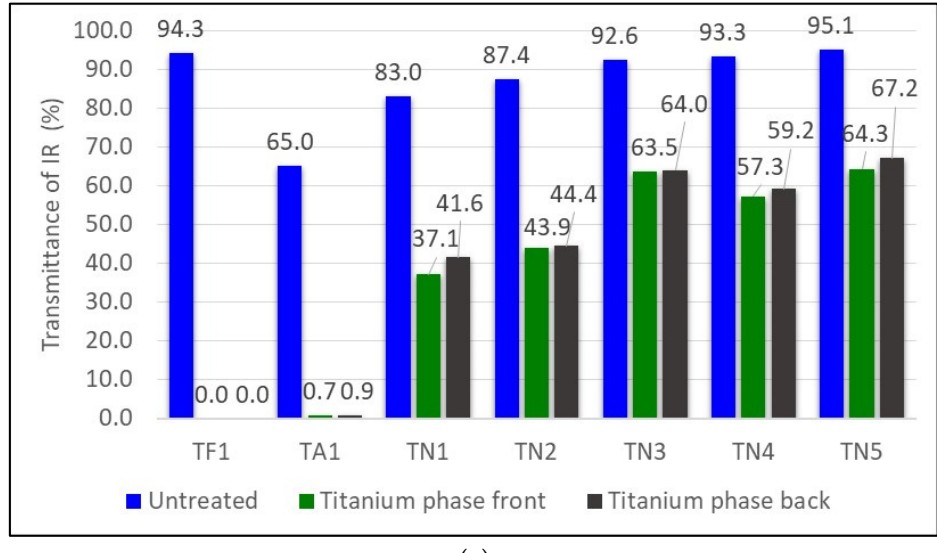

(**a**)

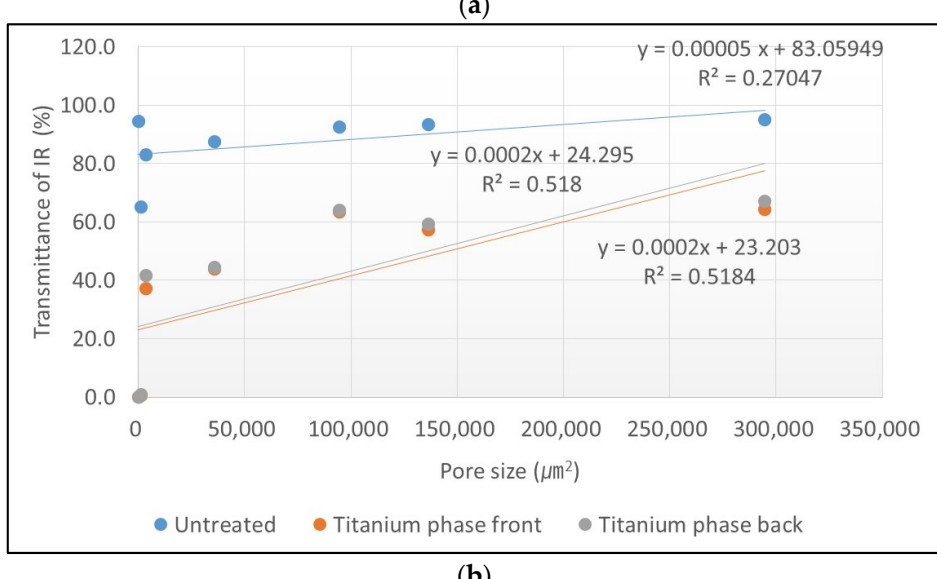

(**b**)

**Figure 6.** Transmittance of IR for each specimen. (**a**) IR transmittance. (**b**) Correlation between IR transmittance and pore size.

### 3.5. Infrared Image Stealth Characteristics According to Heat Transfer at High Temperature

In this research, thermal characteristics attributed to high degree heating sources were examined using IR thermal cameras (Figures 7–10). Photographing was performed with an IR camera with a distance of 0 cm between the sample and the heat source. Only the cross-section was photographed by changing the direction of the titanium sputtering sample. The outward temperature of the heating source was 45.1 to 49.0 °C.

When the titanium layer was facing towards to outside air, the surface temperature was much less than the heating source. When the titanium layer was facing the outside air, the surface temperature was 32.5 °C, and when the titanium layer was directed towards the heating source, the surface temperature was 38.5 °C. However, when the titanium layer faced the heating source, the heating source temperature appeared in the IR image as it was, and there was little hidden effect.

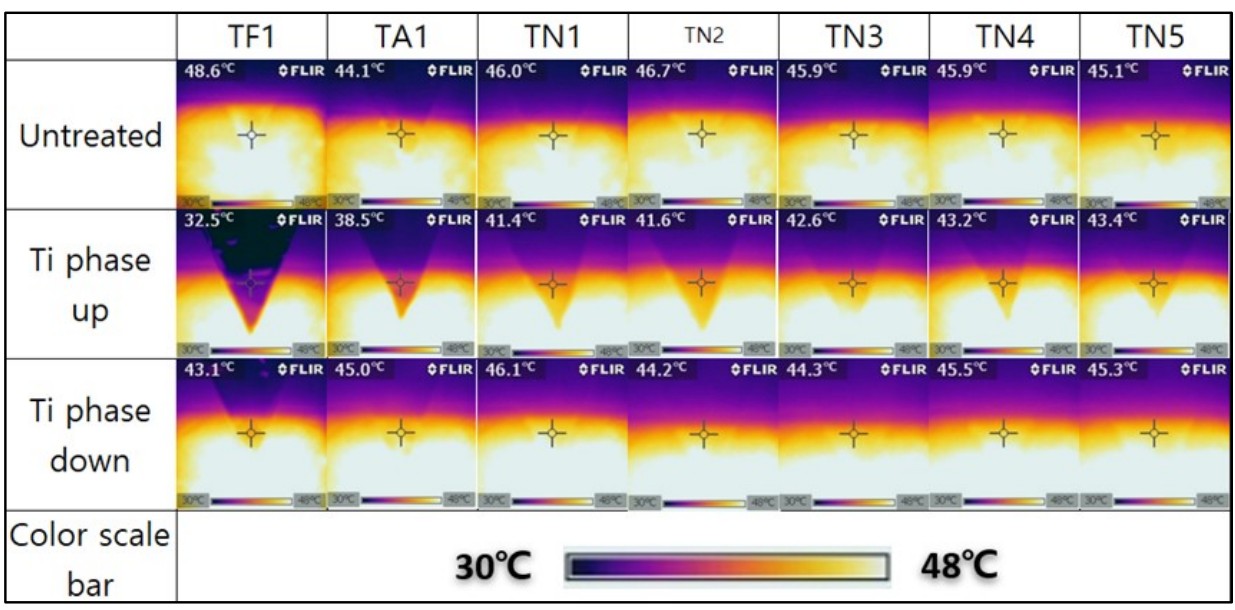

**Figure 7.** IR thermal images according to different densities.

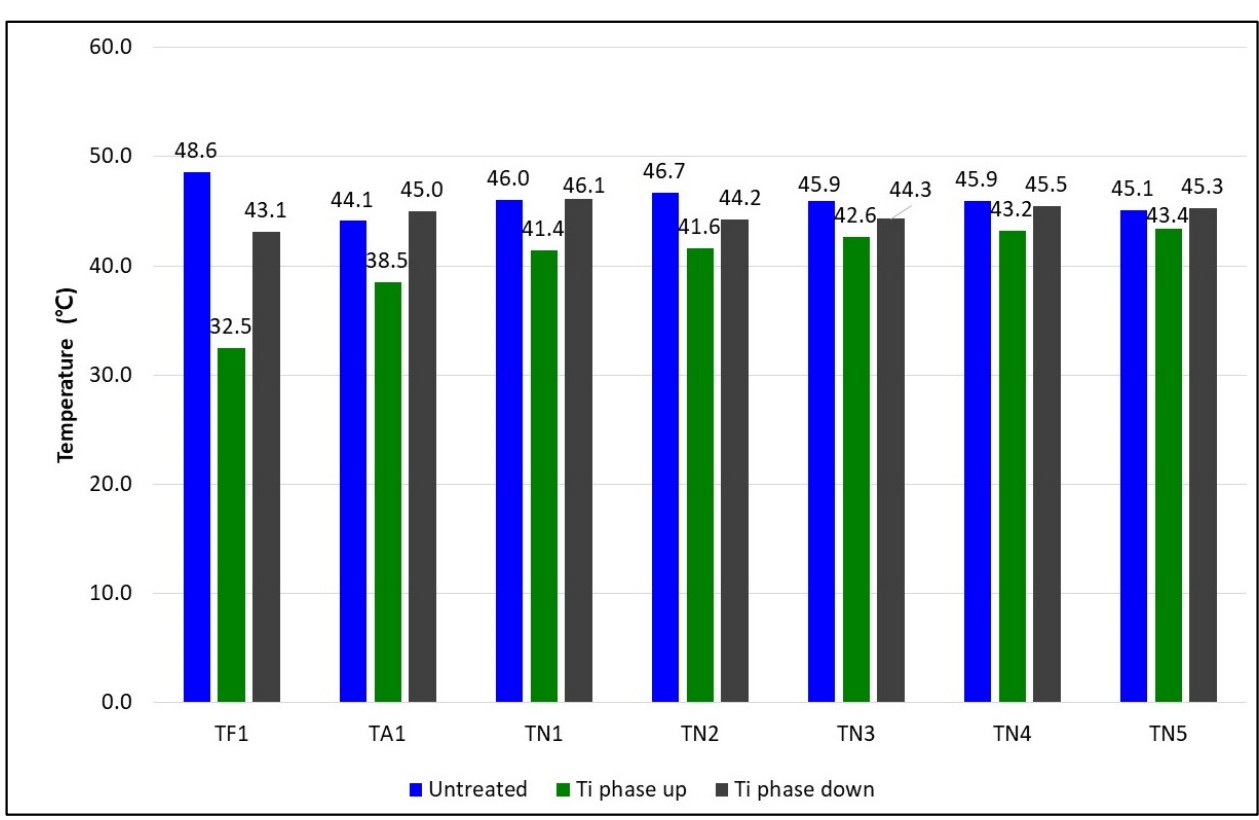

**Figure 8.** Surface temperature with different specimens (distance between heat source and sample: 0 mm).

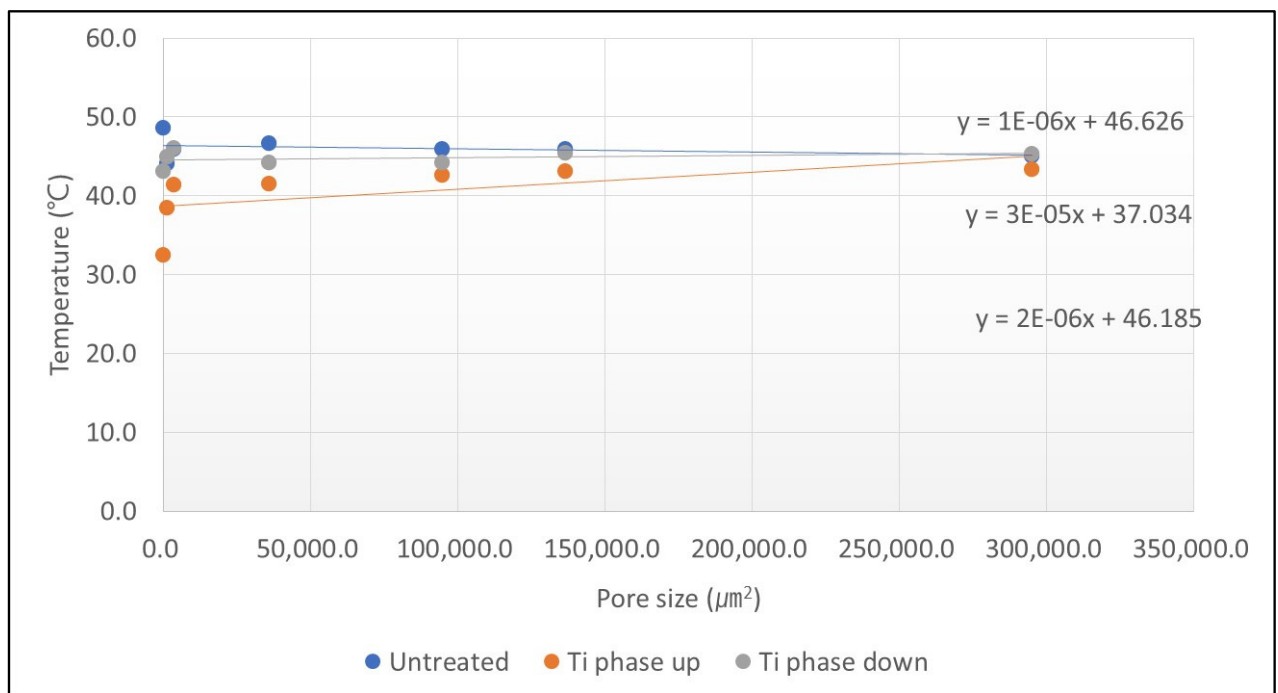

**Figure 9.** Correlation between pore size and surface temperature.

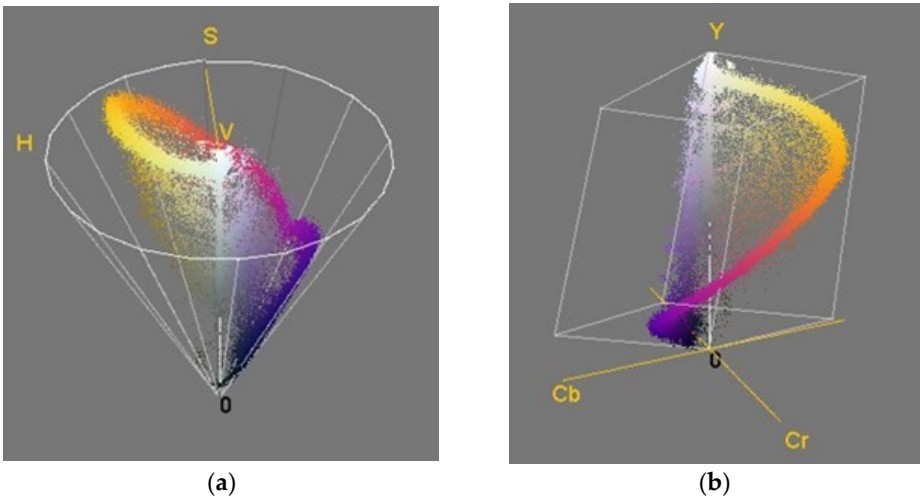

| (a) | (b) |

**Figure 10.** Three-dimensional color distribution. (**a**) Schematic illustration of H, S, and V. (**b**) Schematic illustration of Y, Cb, and Cr.

Moreover, in the case of the mesh specimen, when the titanium sputtering layer was facing the outside air and when the mesh's pore size was larger (3564 → 294,825.5), the surface temperature was from 41.4 to 43.4 °C. When the pore size was bigger, the heat of the heating source was emitted to the outside as it was, and it is believed that the surface temperature was the same as the heating source temperature. In prior studies using aluminum sputtering treatment materials, when the metal layer was directed to the outside air, the surface temperature tended to increase from 27.6 to 34.5 °C as the pore size increased [33].

Furthermore, when the titanium layer of the titanium sputtered mesh sample faces a person's body, the surface temperature was 43.1 to 46.1 °C, indicating a high surface temperature close to the heating source.

These results are judged to be due to the rapid release of heat from the heat source by the titanium layer into the outside air.

After conducting IR imaging, H, S, and V data (Table 3) were obtained using a program (Color Inspector 3D, Image J), and ΔH, ΔV, ΔS, and ΔE data were calculated (Table 4). The measurement Images were shown in Figure 7. The lower right corner of the cross is expressed in Figure 7. And correlation between ΔE and pore size were shown in Figure 11. In addition, the data of ΔH, ΔS, ΔV, and ΔE values are the expressions "Equations (1)–(3)". The H, S, and V data of the outside air were "269, 49, and 53", respectively. Moreover, the H, S, and V data of the heating source were "52, 67, and 93", respectively.

**Table 3.** H, V, and S data of untreated and titanium sputtered polyamide materials.

|  | Untreated | | | Titanium Phase: Up | | | Titanium Phase: Down | | |
|---|---|---|---|---|---|---|---|---|---|
|  | **H** | **S** | **V** | **H** | **S** | **V** | **H** | **S** | **V** |
| TF1 | 72 | 12 | 92 | 280 | 55 | 61 | 57 | 49 | 87 |
| TA1 | 47 | 69 | 93 | 345 | 51 | 79 | 52 | 58 | 91 |
| TN1 | 60 | 28 | 89 | 37 | 93 | 96 | 62 | 34 | 89 |
| TN2 | 60 | 27 | 92 | 39 | 82 | 89 | 55 | 59 | 91 |
| TN3 | 60 | 35 | 90 | 44 | 81 | 92 | 52 | 73 | 92 |
| TN4 | 56 | 38 | 91 | 45 | 85 | 93 | 62 | 27 | 90 |
| TN5 | 49 | 69 | 92 | 47 | 84 | 91 | 57 | 48 | 90 |

**Table 4.** ΔH, ΔV, and ΔS results based on infrared thermal image.

|  | Titanium Phase: Up | | | | Titanium Phase: Down | | | |
|---|---|---|---|---|---|---|---|---|
|  | **ΔH** | **ΔS** | **ΔV** | **ΔE** | **ΔH** | **ΔS** | **ΔV** | **ΔE** |
| TF1 | 208 | 43 | −31 | 214.6 | −15 | 37 | −5 | 40.2 |
| TA1 | 298 | −18 | −14 | 298.9 | 5 | −11 | −2 | 12.2 |
| TN1 | −23 | 65 | 7 | 69.3 | 2 | 6 | 0 | 6.3 |
| TN2 | −21 | 55 | −3 | 58.9 | −5 | 32 | −1 | 32.4 |
| TN3 | −16 | 46 | 2 | 48.7 | −8 | 38 | 2 | 38.9 |
| TN4 | −11 | 47 | 2 | 48.3 | 6 | −11 | −1 | 12.6 |
| TN5 | −2 | 15 | −1 | 15.2 | 8 | −21 | −2 | 22.6 |

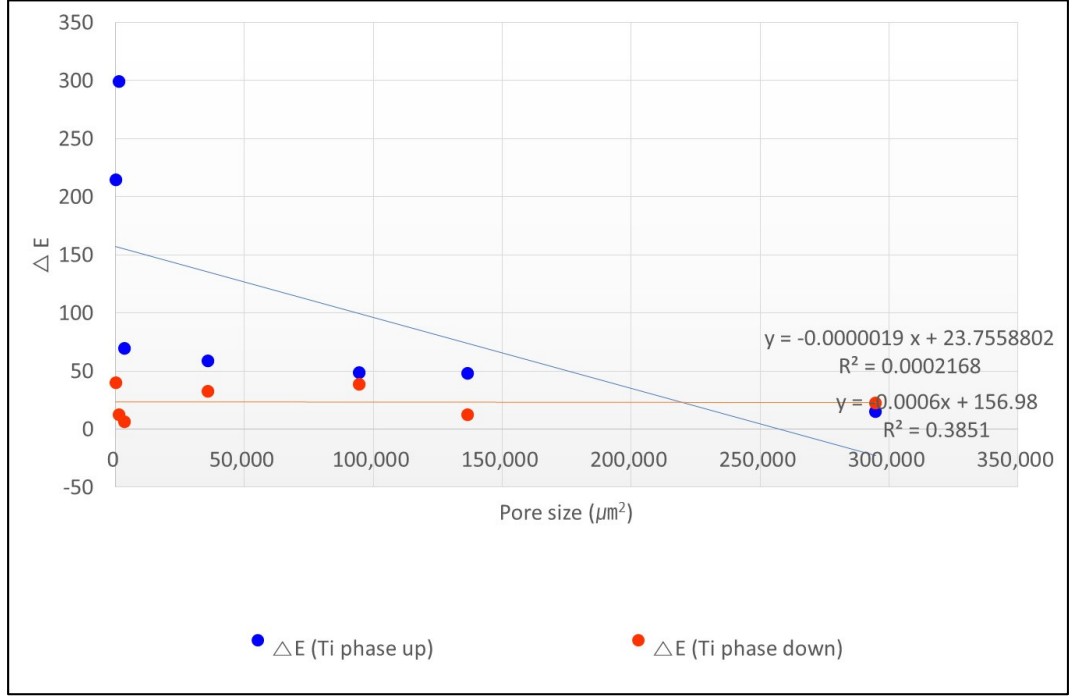

**Figure 11.** Correlation between ΔE and pore size.

The H, S, and V values of "untreated samples" and "Titanium phase down" (when the titanium surface of the cross-sectional sputtered specimen faces the people body) were relatively similar in all specimens. However, the difference in pore size was not significant. H values of all untreated samples were 47 to 72, S values were 12 to 69, and V values were 89 to 92. For all samples of titanium phase down, H values were 52–62, S values were 27–73, and V values were 87–92. It shows a similar aspect to the image of Figure 7. The small absolute value (6.3~40.2) of the specimens of titanium phase down indicates that the difference in H, S, and V data between titanium phase down samples and untreated samples is small. Moreover, in this case, the hidden effect on IR thermal imaging is small. It shows a similar aspect to the thermal image of Figure 7.

On the other hand, the HSV data of "Titanium phase up" (when the titanium surface of the cross-sectional sputtered specimen faces the outside air) represented a different tendency from "untreated sample" and "Titanium phase down". For the data of all specimens in the titanium phase up, the H data were 37–345, the S data were 51–93, and the V data were 61–96. In the H data, the H data tended to decrease as pore size increased. In the case of the V data, they showed a tendency to increase as the pore size of the sample increased. The large amount of absolute data (15.2~298.9) of $\Delta E$ of titanium phase up indicates that the difference in H, S, and V data between titanium phase up samples and untreated samples is large. It can be seen that the titanium phase up sample indicates that there is an alternative hidden effect on the IR imaging.

The Y, Cb, and Cr data (Table 5) of "untreated samples" and "Titanium phase down" were very similar in all samples, regardless of pore size. However, the difference in pore size was not significant. The Y values of all untreated samples were 194–231, Cb data were −13–83, and Cr data were −4–29. For all samples of titanium phase down, Y data were 199–221, Cb data were −42–76, and Cr data were 7–25. The fact that the absolute data of $\Delta Y$, $\Delta Cb$, and $\Delta Cr$ of the specimens of titanium phase down are 5~29, 7~63, and 3~29, respectively, indicates that the difference in Y, Cb, and Cr data between the titanium phase down specimen and the untreated specimen is small, and explains the hidden effect on the IR imaging.

**Table 5.** Y, Cb, and Cr data of untreated and titanium sputter treated specimens.

| | Untreated | | | Titanium Phase: Up | | | Titanium Phase: Down | | |
|---|---|---|---|---|---|---|---|---|---|
| | **Y** | **Cb** | **Cr** | **Y** | **Cb** | **Cr** | **Y** | **Cb** | **Cr** |
| TF1 | 228 | −13 | −4 | 55 | 52 | 35 | 199 | −76 | 25 |
| TA1 | 194 | −83 | 29 | 113 | −11 | 64 | 204 | −68 | 20 |
| TN1 | 217 | −28 | 2 | 166 | −89 | 57 | 212 | −45 | 7 |
| TN2 | 231 | −32 | 7 | 176 | −90 | 49 | 204 | −72 | 19 |
| TN3 | 223 | −35 | 3 | 186 | −94 | 37 | 201 | −72 | 17 |
| TN4 | 212 | −49 | 12 | 181 | −88 | 39 | 221 | −42 | 9 |
| TN5 | 202 | −76 | 27 | 192 | −90 | 32 | 216 | −59 | 12 |

In the case of the Y value, the pore size of the specimen increased as it increased. The absolute data (Table 6) of $\Delta Y$, $\Delta Cb$, and $\Delta Cr$ in the titanium phase up were 10~173, 14~72, and 5~55, respectively. In the case of $\Delta T$ (Figure 12), when pore size was decreased, the larger the amount of absolute data appeared. The large absolute data of $\Delta Y$, $\Delta Cb$, and $\Delta Cr$ indicate that the difference in Y, Cb, and Cr data between titanium phase up samples and untreated specimens are large. This indicates that dense titanium phase-up specimens have an alternative hidden effect on IR imaging.

**Table 6.** ΔY, ΔCr, ΔCb, and ΔT data based on IR thermal images.

| | Titanium Phase: Up | | | | Titanium Phase: Down | | | |
|---|---|---|---|---|---|---|---|---|
| | **ΔY** | **ΔCb** | **ΔCr** | **ΔT** | **ΔY** | **ΔCb** | **ΔCr** | **ΔT** |
| TF1 | −173 | 65 | 39 | 188.9 | −29 | −63 | 29 | 75.2 |
| TA1 | −81 | 72 | 35 | 113.9 | 10 | 15 | −9 | 20.1 |
| TN1 | −51 | −61 | 55 | 96.7 | −5 | −17 | 5 | 18.4 |
| TN2 | −55 | −58 | 42 | 90.3 | −27 | −40 | 12 | 49.7 |
| TN3 | −37 | −59 | 34 | 77.5 | −22 | −37 | 14 | 45.3 |
| TN4 | −31 | −39 | 27 | 56.7 | 9 | 7 | −3 | 11.8 |
| TN5 | −10 | −14 | 5 | 17.9 | 14 | 17 | −15 | 26.6 |

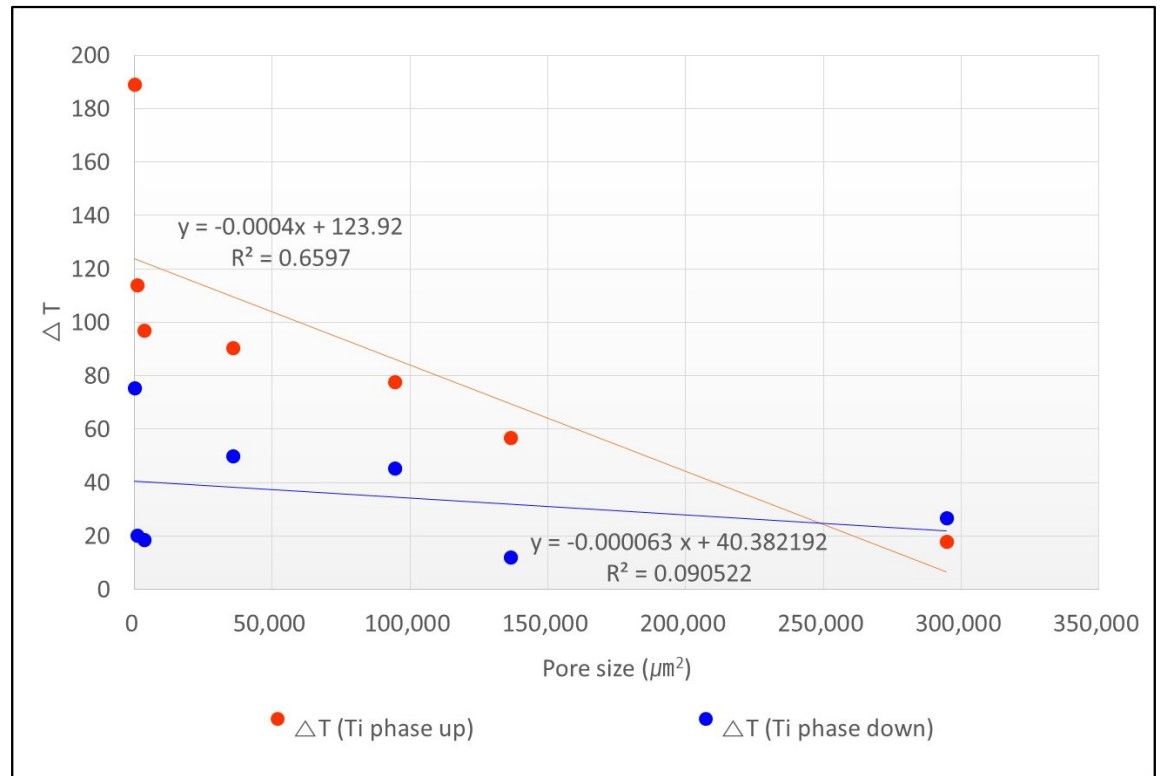

**Figure 12.** Correlation between ΔT and pore size.

## 4. Conclusions

In this study, titanium sputtered specimens were closely observed to determine their electromagnetic blocking properties, hidden effect for IR cameras, electrical characteristics, and thermal properties according to poresize (Table 7). Polyamide specimens with different pore sizes were prepared (film, fabric, and mesh) as base materials for titanium sputtering treatment. Moreover, stealth function for IR thermal imaging was evaluated with H, S, V, Y, Cb, Cr, etc.

As a result of the electrical resistance experiment to observe the electrical conductivity, all untreated specimens had a high amount of resistance data to the extent of overloading, and no electrical conductivity was shown. However, titanium sputtered samples had significantly less electrical resistance data. As a result of infrared transmittance experiments, the infrared transmittance of the titanium sputtered specimens was significantly reduced compared to the untreated specimens. In the results of untreated samples, the transmittance was 95.1 to 65.0%. When the titanium surface faced the IR irradiator and pore size was decreased, the IR transmittance was decreased (64.3 to 0.0%). In addition, when the titanium surface faced the IR measuring device and pore size was decreased, the IR transmittance was decreased (67.2 to 0.0%). That is, the change in the IR transmission data according

to the direction of the titanium sputtered layer was not large. When the pore size of the specimen is larger, the transmittance is higher. In the IR thermal imaging, in the case of the mesh specimen, when the titanium sputtering layer faced the outside air and the mesh's pore size was larger (TN1 to TN5), the surface temperature was 41.4 to 43.4 °C, and hidden characteristics were decreased. As the pore size of the mesh increases, the heat of the human body escapes to the outside air, and the surface temperature is believed to be the same as the heating source temperature. After taking IR thermal imaging, the data of H, S, V, Y, Cb, and Cr were measured to confirm the stealth effect on the quantitative infrared camera, and the values of ΔH, ΔV, ΔS, ΔY, ΔCr, and ΔCb were calculated. The data of H, V, and S of the "untreated sample" and "Titanium phase down" were very similar in all samples, and the difference according to pore size was not significant. In this research, the results indicate that dense titanium phase up samples have an alternative hidden effect on IR cameras. This fact is interpreted as having little influence on the infrared transmittance of the infrared thermal image, and changes in wearing direction, multi-layer manufacturing depending on the purpose of wearing can be considered for compatibility depending on external environmental factors.

**Table 7.** Comparative data according to pore size of sputtered specimens.

| Pore Size of Sputtered Specimens ($\mu m^2$) | Electromagnetic Field of Titanium Phase Up (V/m) | Electrical Resistance (k$\Omega$) | IR Transmittance of Titanium Phase Front (%) | Surface Temperature of Titanium Phase Up (°C) | ΔE Value of Titanium Phase Up | ΔT Value of Titanium Phase Up |
|---|---|---|---|---|---|---|
| 0 | 64.0 | 2.4 | 0 | 32.5 | 214.6 | 188.9 |
| 1200 | 142.0 | 1000.0 | 0.7 | 38.5 | 298.9 | 113.9 |
| 3564 | 143.3 | 5.6 | 37.1 | 41.4 | 69.3 | 96.7 |
| 35,696 | 151.7 | 164.0 | 43.9 | 41.6 | 58.9 | 90.3 |
| 94,440 | 178.3 | 429.5 | 63.5 | 42.6 | 48.7 | 77.5 |
| 136,476 | 165.7 | 1.4 | 57.3 | 43.2 | 48.3 | 56.7 |

The titanium sputtering polyamide samples developed in this study showed excellent electromagnetic wave blocking performance, stealth function for infrared thermal imaging detectors, and IR-blocking characteristics. These findings are expected to be applicable to high-functional clothing, sensors, stage costumes, batteries, and more.

**Funding:** This research received no external funding.

**Institutional Review Board Statement:** Not applicable.

**Informed Consent Statement:** Not applicable.

**Data Availability Statement:** Not applicable.

**Conflicts of Interest:** The author declares no conflict of interest.

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
