# Peer review of "The Contribution of the Pore Size of Titanium DC (Direct Current) Sputtered Condensation Polymer Materials to Electromagnetic Interruption and Thermal Properties"

_coatings, doi:10.3390/coatings13101756_

Round 1

Reviewer 1 Report

In this work, the author investigated electromagnetic wave blocking performance of titanium sputtering treated polyamide materials, hidden effects on infrared thermal imaging cameras, IR transmittance, thermal characteristics and electrical conductivity. Overall, the manuscript is relatively well-organized but has some spelling and formatting errors. This work can be accepted after a major revision.

1.      “In this study, titanium was sputtered into a condensation polymer material and considered in depth in terms of electromagnetic interruption, thermal properties, Infrared blocking and etc.” “Infrared" should be corrected to “infrared”.

2.      There are relevant descriptions in the literatures [29-32] regarding the applied methodologies and the current stage of research results, which should be given in more detail.

3.      Correction to “This research wants to study in depth, Infrastructure function, Infrastructure function, Infrastructure functions, Infrastructure function.” should be made similar to item 1.

4.      Eq. 1 to Eq. 3 should be formatted consistently.

5.      The case of the untreated original samples in Fig. 1 should be given for comparison.

6.      Give the scale of the FESEM images in Figs. 1 and 2.

7.      Give the correct reference format for "Han, 2022".

8.      The sampling process of the cross-section is shown in Fig. 2. Meanwhile, the cross-section images of "TA1" and "TN1" are not vertical with a certain angle of inclination. The measured thicknesses are inaccurate and should be corrected.

9.      “It is believed to be due to the dense intersection of the police station, which has many deadlock points, and the height of the police station furrow is higher than that of the mesh.” Corresponding references should be given.

10.  When the titanium layer was directed to the outside air, the surface temperature was 32.5 ºC, and when the titanium layer was directed to the outside air, the surface temperature was 38.5 ºC.What does this sentence mean? What is the difference between the two cases of 32.5 ºC and 38.5 ºC? The latter seems to the repetition of the former.

11.  “Results and Discussions” and “Conclusions” contained several references to "pores", "the density of the mesh", "the density of the specimen" or "the density of the mesh is loose and the pore size increases". These terms should have the same meaning, so please unify them. Meanwhile, "pores" or "the density of the mesh" should be the key data to influence the experimental results. This data should be given in Table I, not the thickness of the sample.

12.  The conclusion is too long, and should be shortened.

Moderate editing of English language required

Reviewer 2 Report

The authors synthesized titanium DC (direct current) sputtered condensation polymer materials. The test results showed the titanium DC sputtered samples significantly reduced electrical resistance and infrared transmittance compared to the untreated samples. Based on the study results, the authors considered electromagnetic blocking performance, heat transfer characteristics, stealth effects on IR cameras, and the applicability of highly functional smart materials. Although the manuscript is well written with a clear structure and appealing figures, some critical questions are left open. Therefore, this manuscript should be considered for publication only after minor revisions.

(1)   The image is too blurry, please increase the resolution of the image. The font in the image is too small, please revise it. Please add a ruler to the SEM image.

(2)   The line thickness of the tables in the manuscript should be consistent.

(3)   Overall, the manuscript is written well. However, most of the existing literature has not been appropriately referred to substantiate the data provided and inferences drawn.

(4)   Please rewrite the conclusion. The comparative data table can be added to the manuscript.

(5)   How does the pore size of titanium DC (direct current) sputtered condensation polymer materials affect electromagnetic interruption and thermal properties?

The authors synthesized titanium DC (direct current) sputtered condensation polymer materials. The test results showed the titanium DC sputtered samples significantly reduced electrical resistance and infrared transmittance compared to the untreated samples. Based on the study results, the authors considered electromagnetic blocking performance, heat transfer characteristics, stealth effects on IR cameras, and the applicability of highly functional smart materials. Although the manuscript is well written with a clear structure and appealing figures, some critical questions are left open. Therefore, this manuscript should be considered for publication only after minor revisions.

(1)   The image is too blurry, please increase the resolution of the image. The font in the image is too small, please revise it. Please add a ruler to the SEM image.

(2)   The line thickness of the tables in the manuscript should be consistent.

(3)   Overall, the manuscript is written well. However, most of the existing literature has not been appropriately referred to substantiate the data provided and inferences drawn.

(4)   Please rewrite the conclusion. The comparative data table can be added to the manuscript.

(5)   How does the pore size of titanium DC (direct current) sputtered condensation polymer materials affect electromagnetic interruption and thermal properties?

Reviewer 3 Report

  1. Are there any methods in place to fine-tune the pore size since achieving the desired pore size with high precision can be challenging and even slight variations can affect its performance?
  1. How did the authors manage to ensure uniformity in pore size across the entire material? There is no information on the homogeneity or variations in pore size which can lead to uneven electromagnetic and thermal performance, making the material less reliable.
  1. Authors need to provide more details on the sputtering process and subsequent treatments as they may have introduced defects or weakened the structural integrity of the polyamide. It is not clear if the creation of pores is balanced with maintaining the material's strength and durability.

  1. What is the correlation between the pore size and thermal conductivity and/or heat capacity of the material to achieve the desired thermal performance while maintaining structural integrity?
  1. Authors should comment on the compatibility with different environmental factors, like temperature fluctuations, moisture, and chemical exposure, which can affect both its electromagnetic and thermal properties.

  1. The sputtering process can be expensive and time-consuming. Authors need to at least discuss whether the development of their manufacturing process to produce these materials at scale lies within a reasonable cost range.

N/A

Round 2

Reviewer 1 Report

Accept

Minor editing of English language required